# The Role of CCL2/CCR2 Axis in Cerebral Ischemia-Reperfusion Injury and Treatment: From Animal Experiments to Clinical Trials

**DOI:** 10.3390/ijms23073485

**Published:** 2022-03-23

**Authors:** Huixia Geng, Luna Chen, Jing Tang, Yi’ang Chen, Lai Wang

**Affiliations:** 1Institute of Chronic Disease Risks Assessment, School of Nursing and Health Sciences, Henan University, Kaifeng 475004, China; genghuixia@henu.edu.cn (H.G.); lunana@henu.edu.cn (L.C.); 2The School of Life Sciences, Henan University, Kaifeng 475000, China; tangjing@henu.edu.cn (J.T.); 1814030042@vip.henu.edu.cn (Y.C.)

**Keywords:** CCL2, CCR2, cerebral ischemia-reperfusion, neuroinflammation, brain injury

## Abstract

C-C motif chemokine ligand 2 (CCL2) is a member of the monocyte chemokine protein family, which binds to its receptor CCR2 to induce monocyte infiltration and mediate inflammation. The CCL2/CCR2 signaling pathway participates in the transduction of neuroinflammatory information between all types of cells in the central nervous system. Animal studies and clinical trials have shown that CCL2/CCR2 mediate the pathological process of ischemic stroke, and a higher CCL2 level in serum is associated with a higher risk of any form of stroke. In the acute phase of cerebral ischemia-reperfusion, the expression of CCL2/CCR2 is increased in the ischemic penumbra, which promotes neuroinflammation and enhances brain injury. In the later phase, it participates in the migration of neuroblasts to the ischemic area and promotes the recovery of neurological function. *CCL2/CCR2* gene knockout or activity inhibition can reduce the nerve inflammation and brain injury induced by cerebral ischemia-reperfusion, suggesting that the development of drugs regulating the activity of the CCL2/CCR2 signaling pathway could be used to prevent and treat the cell injury in the acute phase and promote the recovery of neurological function in the chronic phase in ischemic stroke patients.

## 1. Introduction

In the inflammatory response, chemokines can regulate the tendency of granulocytic leukocytes to the inflammatory site. According to the difference in amino-acid residues between the first two cysteines, chemokines can be divided into four groups—CXC (α), CC (β), CX3C (δ), and C (γ)—among which the CC subclass is the most abundant [1]. In the central nervous system, both neurons and glial cells express chemokines and their corresponding receptors. In the pathological process of many types of nervous system diseases, an increase in the expression of chemokines and their corresponding receptors in glial cells (i.e., astrocytes, microglia, and oligodendrocytes) and neurons occurs [2]. It has been suggested that the neuron-neuron, neuron-glial, and glial-glial inflammatory signal transduction pathways involving chemokines comprise an important pathological mechanism [3].

Monocyte chemoattractant proteins (MCPs) belong to the CC chemokine sub-family. MCPs, including MCP-1 (CCL2), MCP-2 (CCL8), MCP-3 (CCL7), and MCP-4 (CCL13) in humans and MCP-5 (CCL12) in mice, share similar molecular structural characteristics and chemotaxis of monocytes [4]. Ischemic stroke is a leading cause of disability and death in the world. Genome-wide correlation analyses have shown that high CCL2 levels are closely related to any type of ischemic attack, particularly the large artery and the cardioembolic subtypes [5]. A meta-analysis of 1272 patients with ischemic cerebral infarction and 1210 healthy controls showed that those with gene *CCL2* rs1024611 (-2518A > G) polymorphism had increased incidence of ischemic cerebral infarction, and that the level of serum CCL2 can be used as a biomarker for the early detection and prognosis of ischemic cerebral infarction [6,7,8,9,10].

After cerebral ischemia-reperfusion, monocytes/macrophages are recruited to gather at the ischemic injury site. These cells initiate an inflammatory response during the acute phase to aggravate brain injury but, in the long-term, can promote the migration of neuroblasts and promote the recovery of neurological function.

## 2. CCL2 and CCR2 Signaling Pathway

CCL2, also known as monocyte chemoattractant protein-1 (MCP-1), was the first human monocyte chemokine to be discovered. Its gene is located on human chromosome 17 (chr.17, Q11.2) with three exons and two introns, and it encodes a polypeptide chain with 76 amino-acid residues having a molecular weight of approximately 13 kDa. The molecular structure of CCL2 consists of three distinct domains: a highly flexible N-terminal domain, three anti-parallel β-sheets, and one α-helix. The cysteine residues at the N-terminus of CCL2 form a disulfide bond, giving it a high degree of flexibility. In the primary structure of the CCL2 polypeptide chain, there are three regions that are very important for its biological function—the N-terminal region, amino acids 10–13, and amino acids 34–35—which determine the binding affinity to its receptor and downstream signal transduction. Mutation of these amino-acid residues may lead to a severe decrease in CCL2 chemotaxis ability for monocytes. The expression of CCL2 may be either sustainable or inductive in both physiological and pathological conditions. There are many cytokines or stimulating factors which induce the expression of CCL2, including proinflammatory cytokines (IL-1, IL-4, TNF-α, IFN-γ), macrophage colony-stimulating factors (M-CSF), lipopolysaccharides (LPS), reactive oxygen species (ROS), and oxidized low-density lipoproteins (oxLDLs) [11].

The biological function of CCL2 is realized by binding to its cognate receptor, CC chemokine receptor 2 (CCR2). CCR2 is a seven-time transmembrane G-protein-coupled receptor, which can be divided into two subgroups in humans—CCR2A and CCR2B—and is constitutively or inductively expressed in monocytes, T lymphocytes, or other cells [12]. CCR2 can bind to five proinflammatory cytokines, CCL2, CCL7, CCL8, CCL12, and CCL13, to initiate the downstream signal transduction process, of which CCL2 is the most important ligand [13]. The N-terminal amino-acid residues of the CCL2 polypeptide chain determine the efficiency of its binding to the receptor CCR2 [14]. After CCR2 binds to CCL2, dimerization and internalization occur, which induce monocyte chemotactic protein-1-induced protein-1 (MCPIP1) expression. MCP1P1 initiates the transcription and expression of *interleukin**-1* (*IL-1*), *CCL2*, and *tumor*
*necrosis*
*factor* (*TNF*) genes. Other intracellular protein kinases are also activated by CCR2 binding to CCL2, such as phosphatidylinositol 3-kinase (PI3K), mitogen-activated protein kinase (MAPK), protein kinase C (PKC), nuclear factor kappa-light-chain-enhancer of activated B cells (NF-kB), extracellular signal-regulated kinase (ERK) 1 and 2, Janus kinase (JAK) 2, and JNK1 [15], thus initiating the inflammatory response of the body.

## 3. Expression of CCL2/CCR2 in the Central Nervous System

CCL2/CCR2 is widely expressed in microglia, astrocytes, neurons, and brain microvascular endothelial cells (BMECs) in human and mouse brains [16]. Under normal conditions, the main cortical neurons (Tbr-1 positive) express low CCL2, while the intermediate neurons (CCK/CBR1 interneurons) express high CCL2, suggesting that CCL2 may be involved in the maintenance of emotional state. Meanwhile, in the pathological state, the CCL2/CCR2 signaling pathway formed by the combination of CCL2 synthesized by glial cells and CCR2 on neurons is involved in the formation of neuroinflammation, which may lead to nervous system diseases, such as Alzheimer’s disease (AD), multiple sclerosis (MS), Parkinson’s disease (PD), and so on [17,18].

In the ischemic area induced by cerebral ischemia-reperfusion, the expression of CCL2 is time- and cell-specific. Middle cerebral artery occlusion (MCAO) detected the upregulation of CCL2 in the ischemic area at 12 h, peaking at 2 to 3 days, and then decreasing gradually. The expression of CCL2 in neurons could be detected 12 h after ischemia, while a high expression of CCL2 was detected in astrocytes 2 days after ischemia. This demonstrates the high expression of CCL2 in neurons earlier than that in astrocytes, indicating the special significance of CCL2 in neuronal injury or neuroprotection induced by cerebral ischemia-reperfusion [19].

## 4. CCL2/CCR2 Axis and Cerebral Ischemia-Reperfusion Injury and Treatment

Due to the widespread expression of CCL2/CCR2 in the central nervous system, their expression levels are abnormally elevated when involved in the pathological process of nerve injuries such as stroke, brain trauma, subarachnoid hemorrhage, and psychosis [20,21]. Furthermore, the polymorphism and expression levels of CCL2/CCR2 are associated with the initiation and development process of some central nervous system disorders, such as Alzheimer’s disease (AD), Parkinson’s disease (PD), multiple sclerosis (MS), and schizophrenia [22,23,24,25,26,27,28]. Additionally, CCL2 may be a useful biomarker for early antithrombotic therapy after ischemic stroke [29].

### 4.1. Experimental Research

Stroke is a disease with very high disability and fatality rates, and its pathological mechanism is extremely complex. Experimental studies have shown that the CCL2/CCR2 signaling pathway participates in in the pathological process of cerebral ischemia-reperfusion injury and is closely related to the course of the disease. In the acute phase (<1 day), CCL2/CCR2 signaling is involved in the generation of neuroinflammation and blood-brain barrier breakdown induced by cerebral ischemia-reperfusion, while, in the chronic phase (>1 day), it is beneficial for neuron regeneration.

#### 4.1.1. CCL2/CCR2 Signaling Pathway and Cerebral Ischemia-Reperfusion Injury

After cerebral ischemia-reperfusion, neuroinflammation mediated by the infiltration of inflammatory cells into the ischemic region is one of the important mechanisms leading to brain injury, where the infiltration of inflammatory cells is dependent on the CCL2/CCR2 signaling pathway [30]. Studies in Sprague-Dawley (SD) rats have shown that, after 6 h of ischemia-reperfusion, the expression of CCL2 and CCR2 increased sharply, and the infiltration of monocytes and other inflammatory cells increased, suggesting that they may be involved in the process of neuroinflammatory injury after ischemia-reperfusion [31]; however, *CCR2* gene-specific knockout of monocytes reduced monocyte/macrophage invasion after cerebral ischemia-reperfusion and delayed the neuroinflammatory response by up to 15 days, but also reduced the expression of proangiogenic genes and delayed the recovery of neurological function [32]. These studies suggest the complexity and diversity of the CCL2/CCR2 signaling pathway in the pathological process of cerebral ischemia-reperfusion injury (Table 1).

The mRNA abundance of CCL2 was increased at 6 h, remained elevated at 12 h and 24 h, and lasted for 2 days. This high transcriptional level was maintained at 5 days after ischemia-reperfusion [34,49,50,51]. Consistent with CCL2 mRNA change results, the protein levels of CCL2 were significantly increased 24 h after MCAO [52,53,54]. Mechanism studies have shown that, after cerebral ischemia, secretory phospholipase A2 group X (sPLA2-X) activates lysophosphatidylcholine (LPC) in neurons and astrocytes which, through LPC receptors G protein-coupled receptor 132 (G2A) and P2X purinoreceptor 7 (P2X7R), upregulates the expression of CCL2 and CCR2 in microglia, thus mediating the neuroinflammatory response after cerebral ischemia-reperfusion [35]. Studies on CCL2 transgenic mice have shown that, 24 h and 48 h after MCAO, the cerebral infarction volume of transgenic mice was increased by 67% and 102%, respectively, compared with the wildtype control group, and the concentration of CCL 2 in serum was 40 times higher than that of the wildtype control group [36]. Studies considering a mouse model of middle cerebral artery occlusion and reperfusion have shown that *CCL2* gene knockout decreased the activation of cerebral microglia and the accumulation of phagocytic macrophages, as well as reduced the volume of the cerebral infarction [37].

In the acute phase of stroke (<1 day), the recruitment of monocytes from peripheral blood to ischemic areas is an important step in initiating neuroinflammation; this recruitment process is CCR2-dependent. The levels of CCL2 and CXCL2 were markedly increased in the plasma of mice following MCAO, as well as the microglia and leukocytes infiltrated into the brain [48]. However, for permanent cerebral ischemia, monocyte invasion is CCR2-independent [55,56], suggesting the complexity and diversity of the mechanism of neuroinflammation after ischemic stroke [57]. CCR2^−/−^ or inhibitor (CCR2 pharmacological inhibitor, propagermanium; PG) treatment decreased cerebral infarction volume and mortality at 3 days after stroke; however, after 5 to 28 days, mortality increased, and there was no significant recovery of neurological function, suggesting that CCR2-dependent monocyte/macrophage recruitment is a double-edged sword that aggravates cerebral ischemia-reperfusion injury in the acute phase, but is beneficial to the recovery of neurological function in the long term. In the acute phase, the neuroinflammation mediated by CCL2/CCR2 signaling leads to the rupture of the blood-brain barrier, neuronal apoptosis, and brain damage through the aggravation of proinflammatory macrophage infiltration. In the chronic phase, the CCL2/CCR2 signaling pathway is beneficial to reduce mortality and promotes functional restoration through the promotion of anti-inflammatory macrophage infiltration [56]. The concentration gradient of CCR2 between cortex and choroid is also an important driving mechanism of T-cells invading brain parenchyma after stroke [58]. A study considering a focal transient cerebral ischemia model in mice showed that *CCR2* gene knockout inhibited the increase in blood-brain barrier permeability, decreased the level of inflammatory cytokines, and reduced the volume of cerebral infarction [38]. After MCAO in SD rats, CCL2 can induce human umbilical cord blood cells to migrate to the injured brain region, suggesting that CCL2 not only plays a pathological role in brain injury induced by cerebral ischemia-reperfusion, but also may induce the process of nerve regeneration [59].

How does cerebral ischemia-reperfusion induce the expression of CCL2? A study of cortical slice cultures showed that the excitotoxicity induced by ischemia-reperfusion induces the expression of CCL2 through activation of the NMDA receptor; however, a study on astrocytes showed that the treatment of NMDA does not induce the expression of CCL2, suggesting that there is a heterogeneity of cell type and molecular mechanism, in terms of the high expression of CCL2 induced by cerebral ischemia-reperfusion [60]. In the ischemic area, endothelial cells in blood vessels and peripheral astrocytes express CCL2, thus recruiting CCR2-expressing macrophages to cross the disrupted blood–brain barrier into the brain parenchyma of the ischemic hemisphere, causing a neuroinflammatory response and participating in the brain injury induced by cerebral ischemia-reperfusion. CCL2 is a necessary chemical inducing factor for macrophages and neutrophils to invade the ischemic area, knock out the CCL2 gene, inhibit the invasion of these inflammatory cells after cerebral ischemia-reperfusion, and reduce the volume of cerebral infarction, but it does not affect the activation of microglia in the brain. *CCR2* gene knockout also has a similar neuroprotective effect. A study of CCL2/CCR2 double-knockout mice (CCL2^−/−^/CCR2^−/−^ mice) showed that the expression of other chemokines inducing inflammatory cell invasion was also decreased, such as macrophage inflammatory protein 1 alpha (MIP-1 α/CCL3), CXCL1, C5a, granulocyte colony-stimulating factor (G-CSF), and granulocyte-macrophage colony-stimulating factor (GM-CSF). In addition, the invasion of hematogenous macrophages and neutrophils were significantly inhibited, indicating that the CCL2/CCR2 axis plays an important role in the pathological process of monocytes invading the ischemic area after cerebral ischemia-reperfusion [41]. The stress response after stroke is a common cause of death and recurrence, and the renin–angiotensin system contributes to pressor response after stroke. Studies have shown that CCL2 expressed in the ventrolateral nucleus is involved in the regulation of stress response after stroke by the renin–vasopressin system. CCL2 reduces the expression or activity in the rostral ventrolateral medulla (RVLM), thus relieving the stress response induced by stroke and reducing neuroinflammation [61].

As a dynamic structure, the blood-brain barrier (BBB) strictly restricts the entry of some molecules or cells into the brain parenchyma and maintains the homeostasis of the brain. Brain microvascular endothelial cells (BMECs), pericytes, astrocytes, microglia, and neurons together form the BBB. BMECs connect these cells through a tight junction network to form the primary barrier of the BBB. After cerebral ischemia and reperfusion, CCL2/CCR2 are abundantly expressed in endothelial cells, neurons, microglia, and astrocytes, leading to breakage of the BBB and brain damage [62,63]. A large number of studies have shown that CCL2 is involved in the destruction of the integrity of the blood-brain barrier, especially CCL2, which is removed the C terminus by plasmin processing and has the strongest ability to destroy the blood-brain barrier, thus allowing monocytes to exudate into the brain parenchyma and mediate the neuroinflammatory response [64,65]. Additionally, the effect of CCL2 on BBB leakage is dependent on the CCL2/CCR2 signaling pathway [66].

A study on a blood-brain barrier model of cocultured astrocytes and brain endothelial cells in vitro showed that the high expression of CCL2/CCR2 induced by oxygen-glucose deprivation/reperfusion (OGD/R) increased the permeability to FITC-albumin by 200 times. The treatment with CCL2 antisense RNA and neutralizing antibody decreased the permeability of the blood-brain barrier and increased the relocation of tight junction proteins, while inhibiting CCR2 expression with the same effect [42]. *CCL2* gene knockout decreased the rupture of the blood-brain barrier induced by cerebral ischemia-reperfusion, decreased the expression of the BBB-related genes *occludin*, *zonula occludens-1*, and *zonula occludens-2*, increased the expression of *claudin-5*, and reduced the volume of cerebral infarction [43].

CCL2 destroys the integrity of the BBB mainly through the redistribution of tight junction proteins and reorganization of the actin cytoskeleton. The mechanism of CCL2-induced BBB changes may involve the binding of CCL2 to CCR2 activating Rho-associated kinase or other protein kinases and phosphorylating ezrin/radixin/moesin (ERM) proteins, which bind tight junction proteins and the actin cytoskeleton to destroy BBB integrity [66]. Other studies have shown that CCL2 upregulates aquaporin 4 (AQP4) expression by activating p-p38 MAPK following intracerebral hemorrhage (ICH), causing BBB disruption and brain edema, consequently increasing brain injury [67,68].

#### 4.1.2. The Potential Application of the CCL2/CCR2 Signaling Pathway in the Treatment of Cerebral Ischemia-Reperfusion

Adult neural stem cells have the potential to differentiate into neurons and glial cells, and then migrate to be integrated into existing neuronal circuits. They are expected to replace damaged nerve tissue and restore some brain functions, and they have a wide range of application prospects in the treatment of central nervous system diseases [69].

CCL2, as a marker of adult human multipotent neural cells (ahMNCs), has different expression levels in fetal neural stem cells (fNSCs), mesenchymal stem cells (MSCs), and other stem-cell types involved in tissue repair or regeneration [70,71,72]. After stroke, the migration of neural progenitor cells from the subventricular zone to the injured site to form neural stem cells is the most important strategy to promote the recovery of neural function. The proliferation and migration of neural stem/progenitor cells (NSPCs) in the subventricular zone (SVZ) are CCL2/CCR2-dependent [73,74]. Studies have shown that CCL2 mRNA is upregulated in the ischemic cortex and striatum, and continuously expressed in microglia and astrocytes for 3 days, thus inducing the migration of neuroblasts to the injured area. In this process, CCL2 does not participate in the proliferation and survival of neural progenitor cells, but plays a chemotactic role in neuroblasts [75]. Migratory neuroblasts express a large amount of CCR2; hence, CCL2 or CCR2 gene knockout significantly reduced the migration of neuroblasts to the injured area after cerebral ischemia-reperfusion [76].

After MCAO in SD rats, human umbilical cord blood cells (HUCB) migrated to the brain injury area, and the migration of HUCB was inhibited after treatment with an anti-CCL2 polyclonal antibody, which fully demonstrated the importance of CCL2 in inducing neural stem-cell migration and repairing neural function after cerebral ischemia [59,77]. In the first week after stroke, intraperitoneal injection of CCR2 antibody MC-21 prevented monocytes from recruiting to the ischemic area, thus enhancing nerve regeneration and neurofunctional recovery of the striatum by increasing the survival of newly formed neuroblasts in the subventricular zone (SVZ) and adjacent striatum area [78]. Mesenchymal stem cell (MSC) transplantation is one of the most promising cell-based therapeutic strategies for patients with stroke. After stroke, mesenchymal stem cells are transplanted to improve neurological symptoms and enhance the recovery of neurological function. However, how to ensure the specific migration of mesenchymal stem cells to the ischemic area is the premise of its clinical application. A higher expression level of CCL2 has been found to be beneficial to the migration of mesenchymal stem cells to the cortex, as well as better functional recovery and neurological outcomes after MCAO in rats [79].

In addition, some studies have shown that high expression of CCR2 in mesenchymal cells is conducive to the rapid migration of mesenchymal cells to the cerebral ischemic area, reducing the degree of brain edema and blood-brain barrier destruction, and promoting the recovery of neurological function, suggesting that CCR2 is an important guarantee for mesenchymal cells to target migration to the cerebral ischemic area [80]. CCL2-overexpressing human umbilical-cord-derived MSCs (hUC-MSCs) were intravenously transplanted in rats with middle cerebral arterial occlusion. After 4 weeks, the cerebral infarction volume of the rats in the CCL2-MSC group was much smaller than that in the other groups, as well as neurological function recovery. In addition, the CCL2-MSC group showed increased CCL2 overexpression and migrating stem cells in the peri-infarct regions, accompanied by increased angiogenesis and endogenous neurogenesis, as well as decreased neuroinflammation [81]. Exosome studies from CCR2-overexpressing HUC-MSCs (ExoCCR2) showed that ExoCCR2 could significantly inhibit the migration and activation of macrophages in vitro and in vivo, and it improved post-stroke cognitive impairment (PSCI), oligodendrogenesis, remyelination, and microglia/macrophage polarization [82].

The CCL2/CCR2 signaling pathway plays a decisive role in the chemical tendency of monocytes and T cells in the pathogenesis of chronic inflammatory diseases. Therefore, the gene deletion or inhibitors of this pathway may be used for the prevention of these diseases [83]. The neuroprotection of CCL2/CCR2 signaling inhibitors in cerebral ischemia reperfusion has been confirmed in animal experiments (Figure 1).

Studies on human brain astrocytes have shown that OGD induces upregulation of the expression of CCL2 and CCR2 in human brain astrocytes and inhibits cell growth, while a CCL2 antibody significantly alleviates the inhibition of cell growth. Mechanistic studies have shown that the CCL2/CCR2 signaling pathway may suppress the growth of human brain astrocytes through enhancing the activity of ERK1/2 [85]. However, the use of tPA (tissue-type plasminogen activator) has limitations, such as a narrow time window and intracerebral hemorrhage. After the use of tPA in patients with acute ischemic stroke, circulating neutrophils and T cells increase, exacerbating endothelial injury and promoting brain hemorrhagic effects. Bindarit, a CCL2 inhibitor, prevented the infiltration of lymphocytes and the recruitment of myeloid cells, relieved the hemorrhagic transformation, and promoted the neurological function after tPA thrombolysis in rats [86].

A study of CCL2^-/-^ genotypic mice showed that the cerebral infarction volume induced by MCAO was 29% of that of wildtype mice, the accumulation of phagocytic macrophages in the ischemic area was inhibited, and the expression of inflammatory cytokines IL-1 receptor, IL-6, and IL-1β, and GM-CSF was reduced, suggesting that inhibition of the CCL2 signaling pathway is a potential treatment for reducing infarct volume induced by cerebral ischemia-reperfusion [87]. A study considering a hypoxic-ischemic model of neonatal rats showed that CCR2 knockout enhanced the long-term spatial learning ability of female mice in a T-type water maze, decreased the number of circulating monocytes regardless of sex, and decreased the activation of macrophages and microglia in the brain [44]. *CCR2* gene knockout also reduced the cerebral infarction volume after thrombotic stroke [81]. Ischemic preconditioning is one of the main methods for neuroprotection against stroke, but the mechanism is unknown. Studies have shown that the upregulation of CCL2 may be involved in the stroke preconditioning process [88,89,90].

Irbesartan, a CCR2 antagonist, decreased ischemic cerebral lesion volume after middle cerebral artery occlusion (MCAO) [45]. Propagermanium (PG), a CCR2 inhibitor, by downregulating the expression of signal transducer and activator of transcription 1 (STAT1), inhibited the polarization of microglia to the proinflammatory type, inhibited the expression of inflammatory cytokines, reduced brain edema and infarction volume, and promoted the recovery of neurological function [46]. Progesterone inhibited the expression of CCL2 in endothelial cells after cerebral ischemia-reperfusion, reduced lymphocyte infiltration, maintained the stability and function of cerebral vessels after cerebral ischemia-reperfusion, and then played a neuroprotective role [84]. Therapeutic hypothermia (TH) is a potentially effective way to promote neurological rehabilitation in stroke patients, but the mechanism is not clear. Studies have shown that therapeutic hypothermia reduces the activation of microglia induced by cortical local cerebral ischemia, reduces the expression of CCL2, TNF-α, IL-1β, IL-12, IL-23, and other inflammatory cytokines, and promotes the recovery of nerve function, suggesting that therapeutic hypothermia inhibits the expression of inflammatory cytokines and reduces neuroinflammation, which may be one of its neuroprotective mechanisms [91]. Bindarit, a CCL2 inhibitor, significantly inhibited the inflammatory response, attenuated the neurological impairments, and decreased the ischemic infarct area in MCAO/R rats [92]. Furthermore, both bindarit and minocycline prevented apoptosis and electrophysiological dysfunction in Purkinje cells—the principal neurons and sole outputs of the cerebellar cortex—and consequently alleviated gait abnormality, motor incoordination, and locomotor dysfunction in mice after cerebellar hemorrhage [93]. Prophylactic subacute administration of zinc has shown neuroprotective effects following a transient cerebral hypoxia–ischemia, by reducing lipoperoxidation and cell apoptosis in rats. A body of studies showed that subacute administration of ZnCl_2_ prevented the loss of memory function and maintained neural plasticity by increasing the expression of CCL2/CCR2 in the early and late cortical hippocampus of common carotid artery occlusion (CCAO) in rats [94].

### 4.2. Clinical Studies

Experimental studies have shown that the CCL2/CCR2 signaling pathway is involved in the pathological process of neuroinflammation and blood-brain barrier dysfunction in the acute phase after cerebral ischemia-reperfusion, as well as in the process of nerve regeneration in the chronic phase. Clinical studies have also shown that the level of CCL2 in blood or cerebrospinal fluid is closely related to the clinical symptoms of stroke patients, and it may be used as a biomarker for the prognosis of stroke patients (Table 2).

#### 4.2.1. CCL2/CCR2 Signaling Pathway and Cerebral Ischemia-Reperfusion Injury

Stroke is a severe cerebrovascular disease, with a series of neurological and behavioral symptoms caused by disturbance of cerebral blood flow, leading to higher mortality and disability rates [95]. Stroke is the fifth leading cause of death, according to the American Heart Association (AHA). There are approximately 795,000 new or recurrent stroke patients each year. Stroke can be divided into ischemic stroke and hemorrhagic stroke. The former is caused by a sudden termination of cerebral blood flow, which the latter is caused by bleeding due to blood vessel ruptures. Ischemic stroke accounts for 87% of all stroke patients [96].

However, a population-based prospective cohort showed that a high level of basal CCL2 in serum is highly correlated with the risk of any form of stroke (HR: 1.07 (1.01–1.14)) and is more closely related to a high risk of ischemic stroke (HR: 1.11 (1.02–1.21)); however, the association with hemorrhagic stroke remains unclear (HR: 1.02, (0.82–1.29)), which fully indicates that the high expression of CCL2 is a higher long-term risk factor for cerebrovascular diseases [97]. In patients with ischemic stroke, a sharp increase in the expression of CCL2 can be detected in the cerebrospinal fluid, serum, and extracellular vesicles (EVs), and a high level of CCL2 is associated with poor prognosis of the patients [98,99,100]. However, in patients with acute ischemic stroke over 60 years old, the level of CCL2 in plasma is higher than that in patients ≤60 years old, indicating that CCL2 is involved in the diverse pathological mechanism of ischemic stroke in patients of different ages [101]. Furthermore, the genetic patterns associated with stroke have been associated with polymorphisms in the *CCL2* gene [102].

The levels of CCL2 in sera of patients with acute stroke are increased; thus, it is considered as a marker for the detection of large artery atherosclerosis (LAA) stroke in cryptogenic stroke, which has significant clinical significance [103,104]. In addition, the analysis of cerebrospinal fluid (CSF) in patients with ischemic stroke within 24 h of the onset of neurological symptoms showed that the level of CCL2 protein was much higher than that of normal individuals, suggesting that CCL2-mediated neuroinflammation may be involved in the early process of brain injury in ischemic stroke [105]. Analysis of blood from patients with ischemic stroke showed that CCL2 expression levels were already increased only a few hours after the onset of symptoms. The results of multiple linear regression analysis indicated that the CCL2 level in blood was independently related to clinical outcome scores at specific timepoints [106]. Carotid stenosis (CS) is an important risk factor for ischemic stroke. The plasma CCL2 levels in symptomatic patients with CS were significantly higher than in those with asymptomatic CS [107]. A survey of 115 patients with spontaneous intracerebral hemorrhage (ICH) showed that, after adjusting for age, gender, ICH volume, IVH, infratentorial location, and NIHSS score, elevated CCL2 levels in peripheral blood were associated with a worse 90 day modified Rankin Scale (mRS) 6 h after the onset of symptoms. These results suggest that specific inflammatory factors in the acute phase are associated with poor outcome, and therapeutic strategies targeting this inflammatory response are beneficial for the improvement of clinical symptoms [108]. Malignant middle cerebral artery infarction (MMI) is a very serious type of stroke, with a mortality rate as high as 80%. MMI patients account for up to 10% of all stroke patients. Analysis of inflammatory cytokines in serum of patients with MMI using the Quantibody^®^ Human Cytokine Antibody Array showed that the expression level of CCL2 may serve as a diagnostic biomarker or drug target for malignant middle cerebral artery infarction [109,110].

#### 4.2.2. The Potential Application of the CCL2/CCR2 Signaling Pathway in the Treatment of Cerebral Ischemia-Reperfusion

Intracerebral hemorrhage (ICH) is a type of stroke with a higher mortality rate. A study of specimens from patients with acute ICH showed a dramatic increase in CCL2 expression, leading to increased brain edema in patients. This indicates that CCL2 is the key molecule involved in the pathogenesis of ICH. The results of immunofluorescence staining of human samples showed that CCL2 was colocalized with astrocytes, brain microvascular endothelial cells (BMECs), glial fibrillary acidic protein (GFAP), microglia, and neurons after ICH [32].

After ischemic stroke, if the formation of new blood vessels in the ischemic penumbra is promoted, blood reperfusion can be increased, which is beneficial to reducing the volume of cerebral infarction, neuronal remodeling, prolonging survival times, and functional recovery [111,112]. Angiogenesis is a dynamic process that is induced by the sprouting of endothelial cells (ECs) from pre-existing vessels, thus expanding the pre-existing vascular network [113]. Under a number of physiological and pathological conditions, the endothelial cells degrade the basement membrane, migrate to adjacent tissues, proliferate, and aggregate into tubular shape, thereby forming new blood vessels. Mounting evidence has indicated that many cytokines are (directly or indirectly) involved in the regulation of angiogenesis [114]. A study of outgrowth endothelial cells (OECs) derived from ischemic stroke patients showed that a high expression of CCL2 indicates that a high expression of proangiogenic factors is conducive to the formation of blood vessels [115].

In recent years, the efficacy of traditional Chinese medicine in the prevention and treatment of cardiovascular and cerebrovascular diseases has received extensive attention. Danhong injection (DHI) and Naoxintong capsule (NXT), two Chinese medicines approved by the China Food and Drug Administration, have been used to treat ischemic stroke, the effects of which are involved in regulating chemokine signaling pathways, T-cell receptor signaling pathways, and VEGF signaling pathways to reduce ischemic stroke injury [116]. These studies have suggested that traditional Chinese medicine can reduce neurological damage by promoting angiogenesis in ischemic stroke patients.

## 5. Conclusions and Perspectives

Mounting evidence has suggested that polymorphisms of the *CCL2* gene and higher levels of CCL2 in serum are long-term risk factors for any stroke, and animal experiments have also yielded similar results [117,118]. After cerebral ischemia-reperfusion, the CCL2/CCR2 signaling pathway expressed by cells in the brain regulates the infiltration of inflammatory monocytes into the brain parenchyma, promotes the occurrence and development of neuroinflammation, leads to BBB dysfunction, and enhances brain injury in the acute phase (<1 day). However, the CCL2/CCR2 signaling pathway also promotes the migration of neural progenitor cells to the ischemic area in the later phase (>2 days) of stroke, as well as promotes the recovery of neural function (Figure 2).

A study of patients with acute ischemic stroke (AIS) showed that the level of CCL2 in the plasma of acute ischemic stroke patients aged ≤60 years was lower than that in patients >60 years old [101]. It can be seen, from both animal experiments and clinical data, that the pathological mechanisms of the CCL2/CCR2 signaling pathway involved in stroke are complex and various (Figure 1). However, the use of specific inhibitors inhibiting the activation of the CCL2/CCR2 signaling pathway reduced the neuroinflammation induced by cerebral ischemia-reperfusion. Therefore, there are good reasons to believe that the CCL2/CCR2 signaling pathway could be a potential target for therapeutic intervention in the acute phase of ischemic stroke. With the elucidation of the CCL2/CCR2 signaling pathway in stroke pathological injury and nerve repair mechanisms, as well as the clinical development of specific inhibitors, the appropriate inhibitor treatment timepoint can be selected to prevent and treat brain injury caused by stroke, as well as promote the recovery of neurological function in patients.

## Figures and Tables

**Figure 1 ijms-23-03485-f001:**
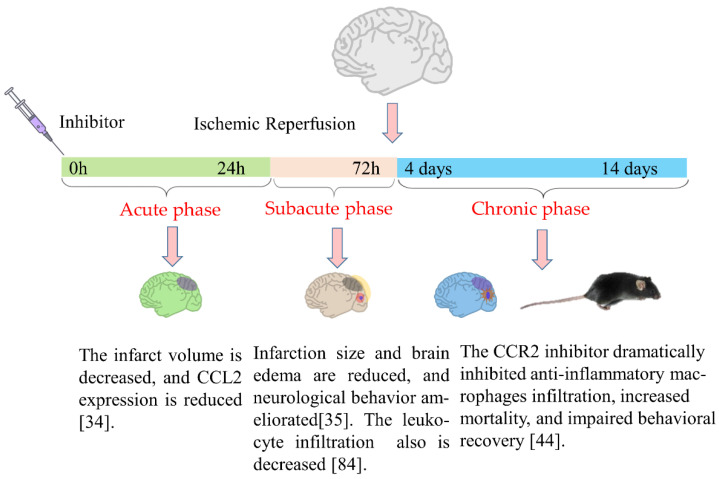
CCL2/CCR2 signaling inhibitors and neuroprotection of cerebral ischemia-reperfusion. In the acute and subacute phases of cerebral ischemia-reperfusion, CCL2/CCR2 signaling inhibitors inhibit the infiltration of proinflammatory leukocytes, reduce brain edema and infarct volume, and promote neurological recovery; meanwhile, in the chronic phase, they inhibit anti-inflammatory macrophage polarization, delay neurological recovery, and increase mortality [34,35,44,84].

**Figure 2 ijms-23-03485-f002:**
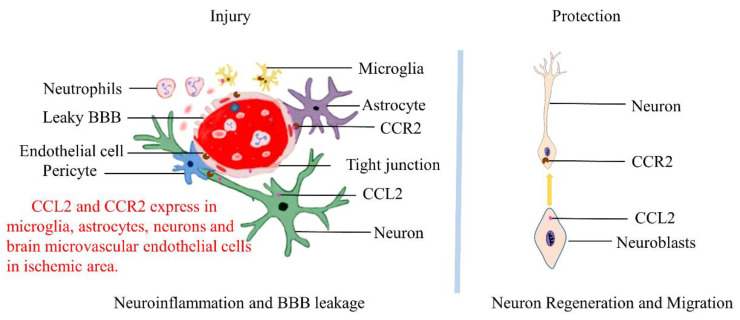
The role of CCL2/CCR2 in cerebral ischemia-reperfusion. The expression of CCL2 and CCR2 in the ischemic area is increased sharply in neurons, astrocytes, microglia, and endothelial cells for 24 h after cerebral ischemia-reperfusion. These induce neuroinflammation and blood-brain barrier leakage, resulting in neurological injury (**Left**). On the other hand, the migration of neuroblasts to the injured area is CCL2/CCR2-dependent (**Right**). As such, the CCL2/CCR2 signaling pathway promotes neuron regeneration and the recovery of neural function after cerebral ischemia.

**Table 1 ijms-23-03485-t001:** The C-C motif chemokine ligand 2/CC chemokine receptor 2 (CCL2/CCR2) signaling pathway and cerebral ischemia-reperfusion.

Number	Species	Animal Model	Gene Type of Animal	Measured Time	Pathological Effects	References
1	SD rats	Middle cerebral artery occlusion (MCAO), 90 min	Wildtype (WT)	24 or 72 h after MCAO	The expression levels of CCL2 and CCR2 were significantly increased, along with microglial activation.	[33]
2	Rats	MCAO, 160 min	Wildtype (WT)	1 h, 6 h, 12 h, and 5 days after MCAO	The CCL2 mRNA expression was significantly increased and remained for 5 days.	[34,35]
3	Mice	MCAO, 120 min	CCL2 transgenic mice	24 to 48 h after MCAO	The infarction volumes were significantly larger in transgenic mice than in wildtype controls, as well as perivascular accumulation of macrophages and neutrophils.	[36]
4	Mice	pMCAO	CCL2-deficient mice	24 h, 36 h, and 2 weeks after MCAO	The infarction volumes, microglia activation, and the accumulation of macrophages were reduced.	[37]
5	Mice	MCAO, 30 min	CCR2^+/+^ and CCR2^−/−^ mice	1 and 5 days after MCAO	CCR2^−/−^ mice had reduced infarct sizes, BBB permeability, and brain edema, compared with CCR2^+/+^ mice.	[38]
6	Wistar rats	MCAO, 90 min	Wildtype (WT)	1, 2, 3, 4, 5, and 7 days after MCAO	The macrophages migrated into the brain parenchyma were dependent on the CCL2/CCR2 signaling pathway.	[39,40]
7	Mice	MCAO, 60 min	CCL2/CCR2 double-deficient mice	1, 2, 4, and 7 days after MCAO	The invasion of macrophages and neutrophils was reduced.	[41]
8	Mice	MCAO, 45 min	Wildtype (WT) and CCL2-deficient mice	12 or 36 h after MCAO	BBB leakage and cerebral infarction were reduced in CCL2-deficient mice.	[42,43]
9	Mice	Hypoxic-ischemic (HI), 40 min	Wildtype (WT) and CCR2 knockout mice	5 weeks and 12 weeks after hypoxia-ischemia	The activation of macrophages and microglia was reduced, along with enhanced long-term spatial learning ability of female mice.	[44]
10	Mice	MCAO	Wildtype (WT)	72 h or 2 weeks after MCAO	CCR2 antagonist inhibited the expression of inflammatory cytokines, reduced brain edema and infarction volume, and promoted the recovery of neurological function.	[45,46]
11	Rat	Transient global ischemia	Wildtype (WT)	8 h and 1, 2, 4, and 7 days after reperfusion	CCL2 is related to the delayed neuronal death in the pyramidal neurons.	[47]
12	Mice	MCAO, 60 min	Wildtype (WT)	23 or 47 h after reperfusion	CCL2 levels were markedly increased in plasma, and microglia and leukocytes infiltrated into the brain.	[48]
13	Mice	MCAO	Wildtype (WT)	48 h after MCAO	The mRNA and protein levels of CCL2 were significantly increased, as well as macrophage infiltration and microglial activation	[49]
14	Mice	MCAO	Wildtype (WT)	6–24 h after MCAO	The mRNA and protein levels of CCL2 were significantly increased 24 h after MCAO.	[50]
15	Mice	MCAO, 60 min	Wildtype (WT)	72 h after MCAO	The expression of CCL2 was significantly increased after MCAO.	[51]

**Table 2 ijms-23-03485-t002:** The CCL2/CCR2 signaling pathway and stroke patients.

Number	Patients	Types of Research	Sample	Cases	Results	References
1	Any stroke	Meta-analysis	Blood	17,180	Higher CCL2 levels were associated with the risk of stroke.	[97]
2	Ischemic stroke	Analytical study	Serum	40	CCL2 levels were increased in ischemic stroke patients.	[98]
3	Acute ischemic stroke	Cohort study	Blood	122	High CCL2 levels were associated with good collateral status.	[99]
4	Hemorrhagic stroke	Comprehensive analysis	Plasma EVs	31	High CCL2 levels correlated with detrimental outcomes after stroke.	[100]
5	Acute ischemic stroke	Pilot study	Plasma		The levels of plasma CCL2 in patients >60 years was higher, compared with patients ≤60 years old.	[101]
6	Ischemic stroke	Analytical study	Blood	71	*CCL2* gene polymorphism was associated with stroke.	[102]
7	Ischemic stroke	Comparative study	Cerebrospinal fluid	23	CCL2-mediated neuroinflammation was involved in the early process of brain injury in ischemic stroke.	[105]
8	Ischemic stroke	Analytical study	Blood	69	CCL2 level was independently related to clinical outcome scores at specific timepoints with ischemic stroke.	[107]
9	Spontaneous intracerebral hemorrhage	Prospective cohort study	Blood	115	CCL2 level was associated with poor outcome in patients with intracerebral hemorrhage.	[108]
10	Malignant middle cerebral artery infarction	Cytokine antibody array	Serums	8	The level of CCL2 may serve as a diagnostic biomarker or drug target for malignant middle cerebral artery infarction.	[109,110]

## Data Availability

Not applicable.

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
