# Peer review of "The Role of CCL2/CCR2 Axis in Cerebral Ischemia-Reperfusion Injury and Treatment: From Animal Experiments to Clinical Trials"

_ijms, 2022, doi:10.3390/ijms23073485_

Round 1
Reviewer 1 Report
The title of the manuscript is completely misleading. Geng et al., should structure their entire review. The authors just describe some of the molecules and signalling pathways involved in CCL2 signalling, however, there is no discussion in the entire review.
The entire text is hard to understand, due to language issues but also due to a very confusing structure (see below)
The manuscript is somehow weird structured. For example:
4. CCL 2/ CCR2 and Cerebral Ischemia-Reperfusion
4.1. Animal Research
4.1.1. MCP-1/ CCR2 Signaling Pathway and Cerebral Ischemia-Reperfusion Injury
4.1.1.1. MCP-1/ CCR2 and Neuroinflammation Induced by Cerebral Ischemia-Reperfusion
there is no text after the headlines - one could have the impression, that the authors just forgot to write something.
Taken all together, the entire review is not bringing any new insights in the topic and is just hard to understand.
Author Response
Thank you for your constructive comments The language issues of the manuscript have been revised by professional language editing. The text after the headlines have been described in this manuscript with red marker.

Reviewer 2 Report
This manuscript addresses a review about the association between CCL2/CCR2 axis in neurological injury. It includes sufficient review about the mechanism and clinical therapeutic potentials of CCL2/CCR2 target for ischemic brain injury with references.
I think a minor revision of manuscript is needed before it can be accepted for publication, which I believe will improve the readability of the paper.
- In section 4.1.1, “MCP1/CCR2” and “CCL2/CCR2” are confusing, so they should be uniformed in the manuscript.
- In the Table1, only clinical research is included, so “Animal Model” should be removed.
- In the section 4.2, also clinical studies should be listed in the table, which is important to know how many/ what kinds of clinical studies are conducted using CCL2/CCR2 pathway targeting.
Author Response
This manuscript addresses a review about the association between CCL2/CCR2 axis in neurological injury. It includes sufficient review about the mechanism and clinical therapeutic potentials of CCL2/CCR2 target for ischemic brain injury with references.
I think a minor revision of manuscript is needed before it can be accepted for publication, which I believe will improve the readability of the paper.
In section 4.1.1, “MCP1/CCR2” and “CCL2/CCR2” are confusing, so they should be uniformed in the manuscript.
Thank you for your constructive comments.
This issue have been revised in this manuscript.
In the Table1, only clinical research is included, so “Animal Model” should be removed.
Thank you for your constructive comments.
This issue have been revised in this manuscript.
In the section 4.2, also clinical studies should be listed in the table, which is important to know how many/ what kinds of clinical studies are conducted using CCL2/CCR2 pathway targeting.
Thank you for your constructive comments.
We have added table 3 in this manuscript, which include clinical studies about CCL2/CCR2 pathway.

Round 2
Reviewer 1 Report
The authors did not really change the style of the manuscript, I do not see a significant improvement.
There are still headlines with no text. The title is still misleading. In this form I think it is not acceptable for publication.
Author Response
The authors did not really change the style of the manuscript, I do not see a significant improvement.
There are still headlines with no text. The title is still misleading. In this form I think it is not acceptable for publication.
Thank you for your constructive comments.
We have revised the title and the style of 4.1 section in this manuscript with red marker.

Round 3
Reviewer 1 Report
the authors improved the writing of the manuscript, it is now easier to follow.
Author Response
the authors improved the writing of the manuscript, it is now easier to follow.
Thank you for your constructive comments.

This manuscript is a resubmission of an earlier submission. The following is a list of the peer review reports and author responses from that submission.
Round 1
Reviewer 1 Report
Huixia Geng et al tried to review the role of the chemokine MCP-1 (or CCL2) and its receptor (CCR2) in cerebral ischemic-reperfusion.
Major comments:
In general the manuscript is poorly structured and poorly written. On the one hand authors try to go much broader in an attempt (but do not succeed) to discuss also the role in Alzheimer's disease, Parkinson's disease, etc. (they even touch SARS-COV-2) and on the other hand they fail to give a good literature summary on cerebral ischemia-reperfusion-induced effects. Sentences are often difficult to understand or are incorrect (also largely due to poor English grammar). The whole text contains a number of repetitions. Titles of paragraphs do not correspond to their content. For instance paragraph 2 is supposed to discuss the "MCP-1 and CCR2 signaling pathway". However, the signaling of CCL2/MCP-1 through CCR2 is not discussed (with the exception of 1 or 2 sentences). The same with paragraph 4.1. Authors indicate at different places in the manuscript (e.g. in lines 175-176) that the CCL2 and CCR2 may be harmful in the initial stages but appear to be essential for tissue regeneration later on. This is interesting but not clearly discussed in one paragraph so that the reader tends to miss this message.
Specific comments:
- Line 26: What do you mean with chemical chemokines? Just chemokines?
- Line 27: What do you mean with quantitative difference? It is just the presence (or absence) of an amino acids that defines the 4 groups of chemokines.
- Line 30-31: neurons and glial cells are mentioned. What about other cells in the brain such as astrocytes?
- Line 38: What do you mean with "MCP has five subfamilies"? Potentially that there are 5 human MCP proteins?
- On line 54 authors claim that CCL2 is a 13 kDa protein. This is not correct. The protein contains 76 amino acids, is smaller than 13 kDa, even if glycosylation (which is not mentioned) would be included.
- On line 57-58 authors state that amino acids 10-13 are important. This is correct but only part of the story. What about other crucial parts of the molecule? At the bottom of that page authors also mention N-terminal amino acids (line 76).
- lines 60-63: these sentences need to be rewritten since the reader can only guess what authors mean.
- line 92-93: "Studies have shown that CCL2 is mainly expressed in astrocytes, while CCR2 is mainly expressed in neurons". I wonder whether this statement is correct? Potentially in a particular conditions but certainly not in general. Ref. 16 discusses Alzheimer's disease in particular.
- line 111-115: Why are these sentences place at the start of this paragraph?
- Line 119-120 is in contrast with the statement on line 121-122.
- Line 216. What is lysine 104? MCP-1 only has 76 amino acids and even with the signal peptide it is only 99 amino acids.
- Line 235-236. How can MCP-1 be different in different cells?
- Line 269-273. What is the meaning of these sentences in this paragraph?
- Line 276. What is IL-1B brain?
- Define all abbreviations at first use.
- Consistently use the systematic chemokine nomenclature (e.g. CCL2 for MCP-1, CCL3 for MIP-1alpha,...). Chemokines are abbreviated CXCL1 (not CXCL-1)
- Tables only refer to part of the existing literature, important manuscripts from several authors are missing. The text should also refer more to the tables.
Reviewer 2 Report
The paper is well written and really interesting. There could be some more discussion on single points (e.g. the chapter with the blood-brain barrier leaves room for interpretation.
What I miss at the end is a final conclusion or summary of their review.
Reviewer 3 Report
The authors summarized the role of monocyte chemoattractant protein-1/CC chemokine receptor 2 (MCP-1/CCR2) in cerebral ischemia-reperfusion, and the data include both animal experiments and human studies. They concluded that this MCP-1/CCR2 pathway promotes monocytes/macrophage recruitment in ischemic stroke and aggravates cerebral ischemia-reperfusion injury in the acute phase, but it is beneficial to the recovery of neurological function for a long term.
It will be conducive for the readers to understand when the inhibitors should be given and when they must be stopped if they can include a figure showing the expression of MCP-1/CCR2 and their effect on neurological function during the different phases of ischemic stroke.
In the conclusions of Table 1, the authors should avoid using “may, might, and should.”
In Table 2, there is a lack of information when the expression level or infarct volume was measured. At acute phase or late phase? One more column should be added to indicate the days after MCAO.
Nerve is the basic unit of the peripheral nervous system. Ischemic stroke causes brain damage that is not a nerve injury. Nerve injury should be changed to brain injury or brain damage. It is common to use the term of chemokines. Chemical chemokine can be changed to chemokine. Ischemic shock in line 41 is the term the readers will get confused. Is that mistyped? The main types of shock include cardiogenic shock, hypovolemic shock, anaphylactic shock, septic shock, and neurogenic shock. What does the ischemic shock mean?
